# Genomic Analysis Reveals New Integrative Conjugal Elements and Transposons in GBS Conferring Antimicrobial Resistance

**DOI:** 10.3390/antibiotics12030544

**Published:** 2023-03-09

**Authors:** Uzma Basit Khan, Edward A. R. Portal, Kirsty Sands, Stephanie Lo, Victoria J. Chalker, Elita Jauneikaite, Owen B. Spiller

**Affiliations:** 1Department of Medical Microbiology, Division of Infection and Immunity, Cardiff University, 6th Floor University Hospital of Wales, Cardiff CF14 4XN, UK; 2Parasites and Microbes Programme, The Wellcome Sanger Institute, Wellcome Genome Campus, Hinxton, Cambridge CB10 1SA, UK; 3Bacterial Reference Department, UK Health Security Agency, London NW9 5DF, UK; 4Department of Biology, Ineos Oxford Institute, University of Oxford, Oxford OX1 3RE, UK; 5NIHR Health Protection Research Unit in Healthcare Associated Infections and Antimicrobial Resistance, Department of Infectious Disease, Imperial College London, London W12 0NN, UK; 6Department of Infectious Disease Epidemiology, School of Public Health, Imperial College London, London W2 1PG, UK

**Keywords:** group B streptococcus, mobile genetic elements, integrative conjugative element (ICE), macrolide resistance, clonal complex (CC)

## Abstract

*Streptococcus agalactiae* or group B streptococcus (GBS) is a leading cause of neonatal sepsis and increasingly found as an invasive pathogen in older patient populations. Beta-lactam antibiotics remain the most effective therapeutic with resistance rarely reported, while the majority of GBS isolates carry the tetracycline resistance gene *tet(M)* in fixed genomic positions amongst five predominant clonal clades. In the UK, GBS resistance to clindamycin and erythromycin has increased from 3% in 1991 to 11.9% (clindamycin) and 20.2% (erythromycin), as reported in this study. Here, a systematic investigation of antimicrobial resistance genomic content sought to fully characterise the associated mobile genetic elements within phenotypically resistant GBS isolates from 193 invasive and non-invasive infections of UK adult patients collected during 2014 and 2015. Resistance to erythromycin and clindamycin was mediated by *erm(A)* (16/193, 8.2%), *erm(B)* (16/193, 8.2%), *mef(A)*/*msr(D)* (10/193, 5.1%), *lsa(C)* (3/193, 1.5%), *lnu(C)* (1/193, 0.5%), and *erm(T)* (1/193, 0.5%) genes. The integrative conjugative elements (ICEs) carrying these genes were occasionally found in combination with high gentamicin resistance mediating genes *aac*(6′)*-aph*(2″), aminoglycoside resistance genes (*ant*(6-Ia), *aph*(3′-III), and/or *aad(E)*), alternative tetracycline resistance genes (*tet(O)* and *tet(S)*), and/or chloramphenicol resistance gene *cat(Q)*, mediating resistance to multiple classes of antibiotics. This study provides evidence of the retention of previously reported ICESag37 (*n* = 4), ICESag236 (*n* = 2), and ICESpy009 (*n* = 3), as well as the definition of sixteen novel ICEs and three novel transposons within the GBS lineage, with no evidence of horizontal transfer.

## 1. Introduction

*Streptococcus agalactiae*, the group B streptococcus (GBS), is a frequent cause of neonatal sepsis. In non-pregnant adults, cases of GBS infections (e.g., bacteraemia) are common in older adults (>75 years), with underlying conditions relative to cases reported in 44–74 year olds [1,2]. Guidelines for first-line therapeutics recommend ampicillin for the treatment of invasive GBS infections in all age groups, with the addition of gentamicin for infants and neonates [3]. Otherwise, penicillin remains the drug of choice for treating GBS infections [4], since GBS is usually susceptible to penicillin. Resistance breakpoints are not available for penicillin in the relevant Clinical Laboratory Standards Institute (CLSI) document [5], while the EUCAST epidemiologic cut-off value for benzylpenicillin has been set at 0.125 mg/L. Multiple emerging reports of GBS with reduced penicillin susceptibility (PRGBS) have been documented globally [6,7,8,9,10,11,12,13,14,15]. Clindamycin and erythromycin were previously uniformly active against GBS; however, an increase in erythromycin-resistant GBS from 3% to 15% and clindamycin-resistant GBS from 3% to 9% was noted in the UK between 1991 and 2010 [16]. Similar increasing trends in macrolide-resistant GBS have been reported in other countries, including Ireland, Italy, China, Portugal, Taiwan, and the USA [4]. In GBS, resistance to macrolides, lincosamides (such as clindamycin), and streptogramin B (MLS*_B_*) is primarily associated with the acquisition of target modification enzymes encoded by *erm* genes, mediating resistance to all three classes or macrolide-efflux pump encoding *mef* genes [17]. The *lsa* and *lnu* genes responsible for targeted lincosamide resistance are observed in GBS intermittently [18]. All resistance genes are associated with mobile genetic elements (MGEs), which are often combined with resistance genes for other antimicrobial classes, such as tetracyclines and aminoglycosides. In addition, they are consolidated as integrative conjugal elements (ICEs) that can move horizontally between bacterial cells, as well as being inherited vertically [17,19,20].

In the present study, we sought to examine the concordance between genotypic and phenotypic antimicrobial resistance profiles of 193 GBS isolates, recovered from adults in the UK between 2014 and 2015. We further analysed whole genome sequences of these GBS to validate the genes conferring resistance and to characterise the MGEs carrying these antibiotic resistance genes.

## 2. Results

### 2.1. Phenotypic and Genotypic AMR Profiling

Seventeen (17/193, 8.8%) GBS isolates carried non-characterised antibiotic resistance genes and were phenotypically susceptible to all eight antimicrobials tested, including tetracycline. Within the 193 GBS isolates from different patients examined, 17 different antibiotic resistance determinants were identified. Of which, 70.5% (136/193) carried resistance genes to only one class of antibiotic and 20.7% (40/193) carried resistance genes to more than one class of antibiotic, i.e., aminoglycosides, macrolides, chloramphenicol, and/or tetracycline. All GBS isolates (*n* = 193) were found to be phenotypically susceptible to ampicillin and vancomycin, consistent with the absence of resistance determinants for these antimicrobials found by in silico genome analysis. All 193 GBS isolates carried penicillin-binding protein (PBP) types (1a, 1b, 2a, 2b, and 2x); therefore, any protein sequence variations identified (Appendix A) were presumed to represent non-resistance polymorphisms concordant with ampicillin susceptible phenotypes. Susceptibility for penicillin was assumed in proxy based on the universal susceptibility to ampicillin.

Established aminoglycoside resistance genes were detected in (8/193, 4.1%) isolates—*aph*(3′-III) and *ant*(6-Ia) (7/8, 87.5%), *aad(E)* (4/8, 50%), and *aac*(6′)*-aph*(2″) (1/8, 12.5%) (Table 1). Phenotypic testing for high gentamicin resistance identified a single isolate (PHEGBS0139) resistant to gentamicin (1/193, 0.5%), consistent with the carriage of *aac*(6′)-*aph*(2″). Resistance to chloramphenicol was found in three isolates (3/193, 1.5%), corresponding to the presence of *cat(Q)* (2/3, 66.6%; PHEGBS0595, PHEGBS0608) (Table 1) or *cat*pC194 (1/3, 33.3%; PHEGBS0738) genes. MIC gradient strips confirmed that all *cat(Q)* (*n* = 2) and *cat*pC194 (*n* = 1) positive isolates had a chloramphenicol MIC of 32 or 48 mg/L. Tetracycline resistance determinants were carried by most GBS isolates (174/193, 90.1%). The most common tetracycline resistance gene was *tet(M)* (154/174, 88.5%), followed by *tet(O)* (23/174, 13.2%), with single isolates carrying *tet(W)*, *tet(S)*, or *tet(L)* (1/174, 0.5%) genes, although *tet(M)* was additionally present for the *tet(L)* and three *tet(O)*-carrying isolates. One of the *tet(M)* positive isolates (PHEGBS0156) remained tetracycline susceptible on phenotypic and MIC testing, with MIC value of 0.06 µg/mL for tetracycline. Genomic analysis for this isolate showed a premature stop codon in the *tet(M)* gene with a 213bp deletion at the 5′ end.

Forty-one (21.2%) GBS isolates showed resistance to macrolides, mediated by at least one macrolide resistance gene, including six isolates (PHEGBS0139, PHEGBS0288, PHEGBS0577, PHEGBS0595, PHEGBS0608, and PHEGBS0071) that harboured multiple macrolide genes (Table 1). The most common macrolide resistance determinant was *erm(A)* or *erm(B)*, distributed equally in (16/41 isolates each, 39%), followed by the identification of *mef(A)* and *msr(D)* in (10/41, 24.3%) isolates, *lsa(C)* in (3/41, 7.3%) isolates, and a single isolate carrying *erm(T)* and *lnu(C)* (1/41, 2.4% each) (Table 1), although five isolates carried combinations of these. Of these genes, *erm(T)* was uniquely carried on plasmid pGB2001 (NZ_JF308630.1), while all other macrolide resistance genes were carried on genomic ICEs. Upon phenotypic testing, resistance rates to erythromycin and clindamycin were 20.2% (39/193) and 11.9% (23/193), respectively, with 10.8% (21/193) resistant to both. Inducible clindamycin resistance was observed in 31.7% (13/41) of macrolide-resistant GBS isolates, representing 12 isolates carrying *erm(A)* and the single isolate carrying *erm(T)*.

Four GBS isolates (PHEGBS0139, PHEGBS0559, PHEGBS0595, and PHEGBS0608) were also resistant to levofloxacin (MIC > 256 mg/L), mediated by double topoisomerase/gyrase point mutations (Ser81Leu for *gyr*A, Ser79Tyr for *par*C), with two isolates (PHEGBS0595 and PHEGBS0608) carrying the H221Y non-resistance polymorphism in *par*E. Twenty-eight additional levofloxacin-susceptible isolates carried the H221Y *par*E substitution, in the absence of *gyrA* and/or *parC* mutations, indicating that this altered amino acid did not contribute to the resistance phenotype. Three isolates (PHEGBS0139, PHEGBS0595, and PHEGBS0608) resistant to levofloxacin were serotype V and ST19, while the other isolate (PHEGBS0559) was composed of serotype III and ST17. Importantly, all four levofloxacin resistant isolates were already multi-drug resistant due to the coincident carriage of ARGs for macrolides (*n* = 4), high gentamicin (*n* = 1), and/or chloramphenicol (*n* = 2) resistance.

### 2.2. Tetracycline Resistance and Mobile Genetic Elements

Tetracycline resistance was most commonly mediated by the *tet(M)* gene (154/174, 88.5%), which was predominantly carried by the transposon Tn*916* (accession no. U09422.1) or Tn*916*-like MGE (87/154, 56.4%), with a lesser number associated with the Tn*5801* MGE (67/154, 43.5%) (accession no. HF930766.1). Among the five common CCs, Tn*916* or Tn*916*-like elements were most common in CC19 (81.2%, 26/32), CC1 (71.8%, 23/32), and CC12 (41.9%, 13/31), whereas Tn*5801* was found in CC23 (89.1%, 41/46), CC17 (58.6%, 17/29), CC452 (100%, 7/7), and CC327 (100%, 3/3). This is consistent with findings of Da Cunha et al. [21]. A novel Tn*916*-like transposon (Tn*7539*; accession no. OP715847) was identified and found to carry the *tet(M)* gene, with the addition of 9 putative additional hypothetical genes, in CC1 (PHEGBS0533) and CC459 (PHEGBS0463) isolates.

Tetracycline resistance was not solely mediated by the *tet(M)* gene: a single novel Tn*916*-like transposon (Tn*7518*; accession no. OP715846) in PHEGBS0586 (CC12, serotype II) was identified, which was found to carry *tet(L)* in combination with *tet(M)*. A novel ICE (ICESag100) was identified to carry the tetracycline resistance gene *tet(W)* alone (isolate PHEGBS0100; ICESag100 accession no. OP715836; Figure 1) inserted at the *rum*A site.

Twenty-three tetracycline-resistant isolates carried *tet(O)* (23/174, 13.2%) and one isolate carried *tet(S)* (PHEGBS0662). Among the 23 *tet(O)* positive isolates, 11 (47.8%) harboured *tet(O)* as a single ARG on an ICE, of which 10/11 (90.9%) carried *tet(O)* inserted into the genome at the 23S rRNA methylase gene (*rum*A), a common target for ICEs carrying serine recombinases [23]. Phylogenetic comparison of *tet(O)*-carrying ICEs indicated two ICE clusters (Appendix A). Cluster 1 was comprised of six closely related ICEs (median SNPs difference of eight SNPs; range 7–70 SNPs) identified in CC12 isolates (PHEGBS0068, PHEGBS0092, PHEGBS0145, PHEGBS0171, PHEGBS0390, and PHEGBS0393), where three elements were identified in serotype II and three in serotype Ib. Cluster 2 was comprised of three moderately related ICEs (median SNPs difference of 312 SNPs; range 207–417 SNPs) (Appendix A), identified in three isolates (PHEGBS0097, PHEGBS0222, and PHEGBS0618). These ICEs were identified in diverse isolates of CC12 serotype Ib, CC19 serotype III, and CC23 serotype Ia. A single ICE carrying *tet(O)* was inserted into a different genomic location into the 50S ribosomal accessory protein L7/12 (*rpl*L) for isolate PHEGBS0132 (serotype III, CC19). Despite the different genomic locations, the *rpl*L *tet(O)* ICE was most similar (82%) to the ICEs identified in isolates PHEGBS0097 and PHEGBS0618. A comparison of the ICE-carrying *tet* genes identified in this study, which were inserted into *rum*A, revealed that the novel *tet(W)* ICE was distinct and did not cluster with any of the *tet(O)* ICEs, with the closest homology of 59.4% to ICESag061 from isolate PHEGBS0061 (Appendix A).

Tetracycline is not used as a therapy for invasive GBS infection. When excluding ICEs that carry only tetracycline resistance, 41/193 (21.2%) carried resistance genes for macrolides and/or lincosamides. One of these detected genes (*erm(T)*) in PHEGBS0539 was carried on a GBS plasmid pGB2001 (NZ_JF308630.1). Figure 2 shows the diversity of CCs and serotypes, which were found to carry macrolide/lincosamide ARGs on ICEs.

In Table 1, it can be observed that eight CC1 isolates and eleven CC19 isolates predominantly carried *erm(A)* or *erm(B)* methylase genes, and to a lesser extent *lnu(C)*, *lsa(C)*, or efflux pump *mef(A)*/*msr(D)* genes. The genomic insertion sites of ICEs carrying these genes widely varied to include Tn*3872*, with no specific insertion site (*n* = 4), *rum*A (*n* = 5), *rpl*L (*n* = 5), *rps*I (*n* = 2), *rum*A + *lsa(C)* (unidentified) (*n* = 1), and two more *lsa(C)*-containing ICEs (undefinable due to the short contig length) (Figure 2 and Table 1). In contrast, the remaining clonal clade CC12 (*n* = 7), CC23 (*n* = 6), CC17 (*n* = 5), CC459 (*n* = 3), and CC452 (*n* = 1) isolates appeared to be less complex and consisted of only 1–2 serotypes and 1–2 ICE insertion sites. CC12, CC23, and CC452 carried single ARG ICEs at a single insertion site. CC17 was comprised of all serotype III and included four isolates carrying the multiple ARG-containing ICESag37 [20] at *rum*A; however, two of these isolates had an additional ARG-containing ICE (ICESpy009-like) inserted at *rpl*L. The final CC17 isolate carried the novel ICESag066 carrying *erm(A)* at the *rum*A insertion site (Figure 2 and Table 1).

### 2.3. Macrolide Resistance and Mobile Genetic Elements

#### 2.3.1. ICEs Carrying the *erm(A)* Gene

Sixteen of the isolates carried the macrolide resistance gene *erm(A)* contained in 12 separate ICEs, of which eight ICEs are novel to this study. The *erm(A)* gene was inserted at *rum*A (*n* = 9), *rpl*L (*n* = 4), *rps*I (*n* = 2), or at an indeterminate site (*n* = 1) (Table 1). Four of these isolates carried a nearly identical ICE, containing both *erm(A)* and *tet(O)*, inserted at *rum*A (representative ICE = ICESag509; accession no. OP715843; Figure 3A). They were initially identified as similar to Sp2905 in *S. pyogenes* (group A streptococcus; accession no. FR691055); however, homology was low (68.2%) and only 26/63 genes (excluding the site-specific serine recombinase) were shared between the ICEs of four GBS isolates and Sp2905. Five additional *erm(A)*-containing novel ICEs were inserted at the *rum*A gene, but lacked the *tet(O)* gene of the previous four isolates: ICESag066 (accession no. OP715837; Figure 3B); ICESag082 (accession no. OP715838; Figure 3C); ICESag098 (accession no. OP715840; Figure 3D); ICESag428 (accession no. OP715841; Figure 3E); and ICESag555 (which is present within ICESag082, highlighted in Figure 3C). ICE size ranged from 12,518 to 126,707 bp and was found in serotype III (CC17, *n* = 1), serotype V (CC1, *n* = 2), and serotype Ib (CC12, *n* = 2). ICESag066 and ICESag555 were nearly identical in size and only differed by 72 SNP, despite being isolated from different serotypes (PHEGBS0066; CC17, serotype III; Figure 3B and PHEGBS0555; CC12, serotype Ib; shown as part of Figure 3C). ICESag082 was found in CC1, serotype V isolate PHEGBS0082 (total length of 79,052 bp; Figure 3C). However, comparing the first 14,150 bp (to the end of the recombinase) shows only 22 SNP relative to ICESag555, despite being from two completely different CC and serotypes. It was observed that the latter was picked up as part of the complex ICE that forms ICESag082.

Four additional *erm(A)*-carrying ICEs were found to be inserted adjacent to the somatic gene *rpl*L, including the previously identified ICESag236 [24] (accession no. OP508058; Figure 4A) in CC19, serotype V isolates, PHEGBS0595 and PHEGBS0608, and those also carried *cat(Q)* and *mef(A)*/*msr(D)* ARGs. Moreover, novel ICESag139 (accession no. OP508059; Figure 4B) was isolated from CC19, serotype V isolate, PHEGBS0139, and was the only ICE to carry the high gentamicin level resistance *aac*(6′)-*aph*(2″) genes in this dataset. Novel ICESag084 (accession no. OP715839; Figure 4C) identified in the CC459, serotype IV isolate, PHEGBS0084, carried *erm(A)* as an isolated ARG. Moreover, a second CC459, serotype IV isolate (PHEGBS0266), carried a single *erm(A)* gene on a sequence contig very short to determine the insertion location (Table 1). The homology between this short contig and ICESag082 was only 68.6%, suggesting that they did not come from a common ancestor ICE. Furthermore, ICE-carrying *erm(A)* was found to be inserted adjacent to the *rps*I (30S ribosomal protein S9) gene (Table 1), a considerably less common somatic insertion site [23]. Isolate 100414 (serotype V, CC1) carried a novel 76-kb ICE- carrying *erm(A)* (ICESag100414; accession no. OP715842; Figure 4D), and a spectinomycin phosphotransferase gene and isolate PHEGBS0044 (serotype Ia, CC23) carried these genes at the same insertion site, with ICE truncated post 55.6 kb, preventing a full ICE comparison. However, the homology between ICESag100414 and the partial ICE for ICESag044 was 27.8%, indicating no common ancestry.

#### 2.3.2. ICEs Carrying the *erm(B)* Gene

Sixteen isolates were found to carry the macrolide resistance gene *erm(B)*, four of which have been discussed above as part of transposon Tn*3875*. For two isolates, sequencing was unable to identify the genomic location due to the short contig length. The remaining 10 *erm(B)*-carrying ICEs were inserted at *rum*A (*n* = 6), *rpl*L (*n* = 3), or *rps*I (*n* = 1) (Table 1). The previously characterised ICESag37 [20], carrying *erm(B)*, *tet(O)*, and three aminoglycoside resistance genes (*ant*(6-Ia), *aph*(3′-III), and *aad(E)*), was found in four isolates (PHEGBS0207, PHEGBS0288, PHEGBS0559, and PHEGBS0577), which were all closely related to serotype III, CC17 with identical ICEs. This suggests retention within lineage for these ICEs, with no evidence of horizontal transfer. The remaining two *rum*A-associated *erm(B)* ICEs, carrying both *tet(O)* and *erm(B)*, lacked aminoglycoside ARGs, and were identified as two novel ICEs: ICESag624 (accession no. OP715844; Figure 5A) identified in PHEGBS0624 (serotype II, CC19) and ICESag071 (accession no. OQ054582; Figure 5B) identified in PHEGBS0071 (serotype III, CC19) (Table 1). These two ICEs were distinctly different from each other and ICESag37, suggesting different evolutionary origins. Moreover, isolates PHEGBS0195 (serotype Ia, CC19) and PHEGBS0206 (serotype IV, CC459) had a combination of *erm(B)* and *tet(O)*, inserted at *rpl*L with two further unique ICEs: ICESag195 (accession no. OP508060; Figure 5C) and ICESag206 (accession no. OP508061; Figure 5D). Furthermore, ICESag662 (accession no. OP508062; Figure 5E) carried *erm(B)*, and was found to be inserted at *rpl*L with *tet(S)*, *ant*(6-Ia), and *aph*(3′-III). The only *erm(B)*-carrying ICE (ICESag738; accession no. OP508063; Figure 5F) inserted at *rps*I was found in isolate PHEGBS0738 (serotype V, CC19), which carried a 19.3-kb ICE in combination with aminoglycoside ARGs *ant*(6-Ia), *aph*(3′-III), and heavy metal resistance genes. This ICE appears to be similar to the *Staphylococcus aureus* composite MGE structure (MES_PM1_) (accession no. AB699882; Appendix A). A second isolate (PHEGBS0156; serotype III, CC19) also had a MES_PM1_ homologous ICE; however, it lacked the IS1618 transposase. Moreover, genomic analysis indicated that the *rps*I region was not disrupted, making it unclear where this last ICE (on a contig devoid of any GBS somatic genes) was inserted (Appendix A). A further difference between PHEGBS0156 and PHEGBS0738 was the latter isolate, which was the only isolate to carry the ARG *cat*(pC-194) inserted near one of the 16S rRNA genes, starting with a site-specific integrase and ending at an IS1216E transposase. The final *erm(B)*-carrying isolate (PHEGBS0622; serotype Ib, CC12) was on a contig very short to clearly define the potential ICE insertion site (Table 1). However, the *rum*A, *rps*I, and *rpl*L sites showed no insertion, and Tn*3872* was not present.

#### 2.3.3. ICEs Carrying *mef(A)*/*msr(D)* Genes

Ten isolates were found to carry the macrolide resistance gene dyad *mef(A)*/*msr(D)* (Table 1). Only a single isolate (PHEGBS0639, serotype V, CC19) was found to carry the macrolide ARG *mef(A)*/*msr(D)* adjacent to the *rum*A gene (Table 1). It was adjacent to the complete genome for *Streptococcus* phage Javan69 (accession no. MK448828.1) and recombinase genes, but the contig was very short to define the genomic location or putative ICE. Five *mef(A)*/*msr(D)*-carrying ICEs were associated in ICEs located at the *rpl*L gene insertion site: Two of these were in combination with *erm(A)* and *cat(Q)*, similar to the ICESag236 shown above (Figure 4A), while the other three *mef(A)*/*msr(D)*-carrying ICEs inserted at *rpl*L had homology to the MGE ICESpy009 (accession no. KU056701) [25]. Two of these isolates (PHEGBS0288 and PHEGBS0577) were serotype III, CC17, and carried the *erm(B)* genes in an ICESag37 inserted at *rum*A (as detailed above), while the other isolate PHEGBS0518 was serotype Ia, CC452 (Table 1). Four of the remaining *mef(A)*/*msr(D*)-carrying isolates were serotype Ia and CC23 (PHEGBS0067, PHEGBS0070, PHEGBS0248, and PHEGBS0625) and the *mef(A)*/*msr(D)* genes were located adjacent to a *tet(M)*/Tn*5801* cassette with no clear MGE. The final *mef(A)*/*msr(D)*-carrying isolate near to, but more distant from, a Tn*5801* cassette was of a different serotype (PHEGBS0625; ST23, serotype Ia), with no clear MGE (Table 1).

#### 2.3.4. ICEs Carrying the *lsa(C)* Gene

Lincosamide-resistance genes were also identified. A single isolate (PHEGBS0139) carried the *lnu(C)* gene on a 1724-bp region, including an IS1 transposase homologue and bounded by 25-bp imperfect inverted repeats flanked by 8-bp direct repeats, as reported previously for another GBS isolate [26]. However, this isolate also carried a unique *erm(B)*-carrying ICE (ICESag139) inserted at the *rpl*L site, making the *lnu(C)* gene redundant, as discussed above (Figure 4B). Three isolates were found to carry *lsa(C)* genes: these were CC19 isolates and despite being different serotypes (PHEGBS0222; serotype Ib and PHEGBS0372; serotype III), the *lsa(C)* insertion site has been previously described as ICE2603 tRNA^lys^ element [27]. In the third CC19 isolate (PHEGBS0071; serotype III), the *lsa(C)* gene appears to have been incorporated into a *tet(M)*-positive tn*916*-like MGE (novel Tn*7589*; accession no. OQ269476), which has been inserted at the stop codon of an TVP38/TMEM64 family protein gene. All these isolates yielded intermediate resistance to clindamycin, except for PHEGBS0071, where the high level clindamycin resistance is likely mediated by *erm(B)* found on ICESag071 (Figure 5B), obscuring the activity of *lsa(C)* for this isolate.

## 3. Discussion

Overall, a high concordance rate (99%) between phenotypic and in silico predicted genotypic antibiotic resistance profiles were found. No phenotypic or genotypic resistance to the current first line antibiotic (penicillin) was detected in this study; therefore, all variations in the penicillin binding protein (1a, 1b, 2a, 2b, and 2x) are deemed to be non-resistance mediating polymorphisms, consistent with the theory that resistance to penicillin is rare in GBS [4,5]. However, there are growing reports of isolates with PRGBS in some geographical regions, including two isolates identified to date in Hong Kong [6], seventy-five in Japan [7,8,28], fifty-two in the USA [10,11,12], two in Canada [13,14], seven in Mozambique [15], and two in South Africa [29] and Korea [30]. However, the resistance rates for second-line antibiotics, such as erythromycin and clindamycin, continue to increase internationally, to the point where current recommendations for second-line therapeutics for adults have changed to cephalosporins or vancomycin, in response to the detailed BMJ best practice guidelines [3].

To treat severe invasive GBS infections, such as infective endocarditis, periprosthetic joint infections, or even early-onset disease (EOD) in babies [3,4], combination therapy that includes penicillin and gentamicin has been suggested. The first case of GBS showing high-level resistance to gentamicin was observed in a strain isolated in 1987 from a French patient [31]. The mechanism of acquisition of high-level gentamicin resistance determinants in GBS is largely unknown. We identified a single isolate in this study with high level gentamicin resistance. Taking into account the emerging resistance trends, there is a need to consider other classes of antibiotics with therapeutic potential for GBS infections.

Tetracycline is no longer a therapeutic option for GBS, since the extensive use after its first discovery in 1948 led to the high resistance profile in many bacteria, including GBS. Da Cunha et al. speculated that the extensive use of tetracycline in the 1960s exerted selective pressure on GBS, leading to genetic diversity loss and emergence of few tetracycline resistant pathogenic clones, which also carried mobile genetic elements with a mixture of ARGs, including tetracycline resistance [21]. As expected, in our study, the rates of resistance to tetracycline were high (174/193, 90.1%), largely mediated by *tet(M)* (154/174, 88.5%) and predominantly carried by two related ICEs: Tn*916*/Tn*916*-like elements (87/154, 56.4%) and Tn*5801* [32] (67/154, 43.5%), which were found to be prevalent in CC1, CC12, CC19, CC17, and CC23, respectively. Similar observations were seen in previous studies [21,33], suggesting that these ICE-harbouring *tet(M)* are clonally related and acquired through limited and rare insertion events, followed by expansion within lineages. While these MGEs are generally accepted as promiscuous, Da Cunha et al. [21] previously demonstrated that five of the major clonal complex clades have fixed genomic insertion sites, resulting from the high level of tetracycline use in the 1960s, followed by expansion within lineages. This includes the *erm(B)*-carrying variant referred to as Tn*3872* (CZ-NI-013 strain ENA, accession no. ERR048605), which Da Cunha et al. found clustered in CC1. Moreover, this is consistent with our finding that only CC1 isolates carried *erm(B)* in Tn*3872* (accession no. OP715845).

We also found four isolates resistant to levofloxacin (fluoroquinolone) due to double point mutations in *gyr*A and *par*C. Interestingly, all four isolates containing quinolone resistance determinants belonged to CC19, and were resistant to antibiotics from at least two other classes, as seen previously [34].

Previous studies have examined the correlation between serotypes and clonal complexes, relative to antimicrobial resistance or the presence of ARG classes. Some studies suggest that erythromycin resistance is more associated with serotype V [35,36] or CC12 and CC17 [37], which is not supported in our data. However, all studies examining resistance trends agree that increases in macrolide and lincosamide resistance are fairly universal [38]. In the UK, the reported increase for erythromycin (3% to 15%) and clindamycin resistance (3% to 9%) noted from 1991 to 2010 [16] has continued, with recent reports indicating 31% clindamycin and 36% erythromycin resistance in 2020 (Laboratory surveillance of pyogenic and non-pyogenic streptococcal bacteremia in England: 2020 update (publishing.service.gov.uk). In this study, the data from 2014 to 2015 are consistent with this trend for both clindamycin (11.9%) and erythromycin (20.2%). Further in-depth investigations of larger GBS datasets are needed, since this limited study indicates that no single ARG drives the increase in resistance. While 32 of 193 isolates carried *erm(A)* or *erm(B)* genes, they were carried on 16 unique ICE elements (Table 1); five isolates had contigs very short to clearly define potential ICEs (Table 1). A maximum of four isolates carried the same ICE element (Tn*3872*, ICESag37, and ICESp2905-like) and in each case, they were all found in the same CC (Table 1), as observed previously for Tn*3872* in CC1 [21] and ICESag37 in CC17 [20], suggesting their association, spread, and maintenance.

We also identified three hotspot integration sites for ICE-carrying macrolide resistance determinants, which includes the most frequent *rum*A found for sixteen ICEs, *rpl*L for ten ICEs, and less commonly for *rps*I ICEs (*n* = 3) (Figure 1 and Table 1). These integration sites have previously been classified as well-known integration sites for ICEs in streptococci [23]. Additionally, sixteen novel ICEs carrying macrolide alone or with other antibiotic resistance determinants (ICESag066, ICESag071, ICESag082, ICESag084, ICESag098, ICESag139, ICESag195, ICESag206, ICESag236, ICESag428, ICESag509, ICESag555 (separate as well as part of ICESag082), ICESag624, ICESag662, ICESag738, and ICESag100414), a single novel ICE carrying the tetracycline resistance gene *tet(W)* alone (ICESag100), and three new transposons (i.e., Tn*7518* carrying a combination of *tet(M)* and *tet(L)* genes, Tn*7589* carrying *tet(M)* and *lsa(C)* genes, and Tn*7539* carrying typical Tn*916* with an additional nine ORFs) were identified.

The number of novel ICEs identified in this small dataset (only 193 GBS isolates from the UK) is surprising, and confirms that the GBS population is highly diverse. This indicates that specific antimicrobial resistance patterns conferred via MGEs could potentially remain within specific lineages and expand or emerge, rather than spread via the horizontal transfer. Therefore, the proposed GBS vaccines would not only potentially eliminate the most common serotypes/genotypes of disease-causing GBS, but also could be used to limit the emergence and spread of antimicrobial-resistant and multi-drug-resistant GBS lineages. To better understand how vaccines and changes in antimicrobial prescriptions for the treatment of GBS infections would impact antimicrobial resistance, global longitudinal molecular epidemiology studies of GBS carriage and disease are required. Moreover, this will provide details on the molecular basis of antibiotic resistance.

## 4. Materials and Methods

### 4.1. Clinical GBS Strains and Their Whole Genome Sequences

The 193 invasive and non-invasive GBS isolates collected from adults in the UK between 2014 and 2015, as analysed in our previous study [39], were used in this study.

### 4.2. Phenotypic Antibiotic Susceptibility Testing (AST)

All susceptibility testing was performed on pure archived GBS strains, recovered from freezing by overnight culture on blood agar supplemented with 5% sheep blood (Oxoid, Basingstoke, UK). Then, overnight bacteria inocula were resuspended in sterile saline to 0.5 McFarland units, containing 10^5^ cfu/mL. Disk diffusion and gradient strip AST were performed on Mueller Hinton with 5% sheep blood (MH-F plates; Oxoid, Basingstoke, UK). In addition, uniform lawns of freshly-made isolate suspensions were inoculated prior to overlaying with antimicrobial impregnated disks or gradient strips (Oxoid), after visible liquid had dried from the plate surface. A disk dispenser (Thermo Fisher Ltd., Loughborough, UK) was used to ensure uniform spacing of 1 cm between disks on the 90 mm plates. Then, the plates were incubated in an aerobic incubator at 35 °C for 24 h.

All 193 isolates were tested for antimicrobial susceptibility by disk diffusion using the following concentrations: Ampicillin (10 µg), vancomycin (30 µg), tetracycline (30 µg), erythromycin (15 µg), clindamycin (2 µg), chloramphenicol (30 µg), high level gentamicin (120 µg), and levofloxacin (5 µg) (Oxoid, Basinstoke, UK). Briefly, the inhibition zone sizes were recorded for each antimicrobial agent and interpreted according to the Clinical Laboratory Standards Institute (CLSI) 2022 guidelines [5]. For this method, erythromycin and clindamycin disks were adjacent and 1 cm apart to identify clindamycin resistance induction via the D-zone test, as previously described [5,40]. Briefly, blunting (or D appearance) or bacterial growth within the clindamycin zone of inhibition proximal to the erythromycin disk was documented as inducible resistance, commonly mediated by MLS_B_-inducible methylation. In the case where the zone of inhibition did not clearly define the sensitivity of an isolate, or where genotypic evaluation of an isolate did not match the expected phenotypic profile (except for tetracycline), gradient strips (MIC test strip; Liofilchem, Italy) were employed to determine a more accurate MIC on MH-F plates, as prepared above for disk diffusion. The MIC value, where the growth of the GBS touched the edge of the strip, was interpreted according to the CLSI 2017 guidelines [5].

### 4.3. Long-Read Whole Genome Sequencing

Ten isolates (study accession no. PRJEB59039) were selected for additional long read sequencing, guided by the initial short read bioinformatics analysis. The gDNA was prepared as previously described [41] using the Qiagen QIAamp DNA kit and extracted on the QIAcube platform (Qiagen, Manchester, UK). Moreover, the gDNA was quantified using the dsDNA HS assay kit and the Qubit 4.0 fluorometer (Thermo Fisher Scientific Inc., Glasgow, UK). Then, it was purified and concentrated using SPRI beads (Mag-Bind TotalPure, Omega Bio-tek, Inc., Norcross, GA, USA) at a 1:1 ratio with a final elution volume of 15 µL, in order to achieve an optimal range between 40 and 60 ng/µL. Moreover, the gDNA was re-quantified with the dsDNA BR assay kit. Genomic libraries were prepared using the Rapid Barcoding Kit (SQK-RBK004; Oxford Nanopore; Oxford, UK), sequenced on a R9.4 flow cell using a MinION (Oxford Nanopore Technologies; Oxford, UK), and base-calling was performed using Guppy v4.0.9 within MinKNOW v20.06.4. Long reads were demultiplexed using porechop (v0.2.4) [28], and assembled against corresponding short reads generated from the Illumina MiSeq using Unicycler (v0.4.7) [42] with default parameters. The hybrid assembly was assessed using Quast (v5.0.2), and ABRicate (v0.9.7) (>98% coverage and identity) was used to detect antimicrobial resistance genes (ARG). The contig containing the ARG of interest was exported to a separate FASTA file for comparative analysis using Bandage (v0.8.1), in order to accurately determine the boundaries of the ICE carrying the ARGs, or to accurately determine the full novel ICE composition where ARGs were not bounded by somatic genes on both sides in short-read contigs [43]. Novel transposons were assigned Tn numbers using the Transposon registry [44].

### 4.4. Resistome Profile Associated with the Mobile Genetic Elements

Clonal complexes (CCs) were assigned by PubMLST using *S. agalactiae* MLST database (https://pubmlst.org/sagalactiae/, accessed on 15 December 2022). AMR genes were identified by ResFinder 2.1 database/webserver [45]. Assembled GBS genomes were screened for MGEs using Geneious Prime^®^ (Biomatters, Inc., Auckland, New Zealand) by annotation (threshold 97% identity) with genes identified by ResFinder. Genomic regions containing MGE became apparent when assemblies were annotated using the prototype genome FDAARGOS_520 (accession no. CP033810), which was selected as it did not have any evidence of ICEs. When a second round of annotation was performed using a database of existing ResFinder resistance genes, the presence of ARG annotations within these gaps identified the MGEs. These MGE insertions usually corresponded to previously reported *Streptococci* insertion “hotspots” [32], i.e., bounded on one end by *rum*A (23S rRNA (uracil-5) methyltransferase; protein_id: AQY23883.1, *rpl*L (50S ribosomal accessory protein L7/12; protein_id: AWZ30204.1), or *rps*I (30S ribosomal protein S9; protein_id: AKI94631.1), which were confirmed by the alignment of these references to query assembled genomes in Geneious Prime. BLASTn searches against NCBI databases in Geneious Prime^®^ were carried out as the final step in identifying and annotating the MGE-associated genes present between DRs of the putative ICEs in each genome. Artemis [46] was used to colour the detected signature proteins of an ICE, such as tyrosine/serine integrases/recombinases, relaxases, plasmid mobilisation protein genes (*mob*A/*mob*L), conjugal transfer protein genes (*vir*B4, *vir*D4, *tra*L, and *trb*L), and insertion sequences. Bakta v1.5 was used to annotate putative MGEs (https://github.com/oschwengers/bakta, accessed on 15 December 2022). Pairwise BLASTn alignment of identified MGEs was performed and visualised using Easyfig v2.2.2.

### 4.5. Phylogenetic Analysis

Neighbour joining method (using Jukes-Cantor genetic distance model, global alignment with free end gaps, and cost matrix of 65% similarity) was used to construct the phylogenetic tree for ICEs carrying the *tet(O)* gene alone with an ICE-carrying *tet(W)*, which was used as an outgroup in Geneious Prime^®^ (Biomatters Ltd., Auckland, New Zealand) tree builder tool.

## 5. Conclusions

This study defines sixteen novel ICEs and three novel transposons, with no evidence of horizontal transfer. In addition, it provides evidence of retention within the GBS lineage of previously reported ICEs. The discovery of numerous novel MGE-carrying ARGs in a small number of isolates reflects the importance of ARG study in human pathogens, in which resistance is of increasing concern. It was surprising to identify numerous novel ICEs present in a small cohort. The GenBank reference genomes for the mobile elements discovered here will be of benefit to future studies in determining whether increasing antimicrobial resistance is a result of global spread of the ICE defined here, or whether they are being caused by the sporadic capture of other novel ICEs. The ICE sequences and genomic dataset provided by this and future studies enable the development of tests to detect genes encoding resistance, as well as the improvement of the surveillance and monitoring of resistance potential in the population. Moreover, it could aid in an increasingly rapid therapeutic intervention than sensitivity testing alone.

## Figures and Tables

**Figure 1 antibiotics-12-00544-f001:**
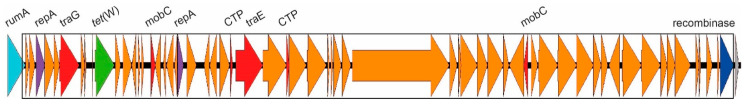
A novel 63.9-kb ICESag100 carrying the *tet(W)* gene (green) inserted at the *rum*A site (turquoise). Arrows indicate mobility genes and conjugal transfer proteins (red), *rep*A initiator genes (purple), critical site-specific recombinase (blue), and other ICE genes (orange) relative to somatic post-insertion genes (grey). Gene orientation is depicted by the arrow direction.

**Figure 2 antibiotics-12-00544-f002:**
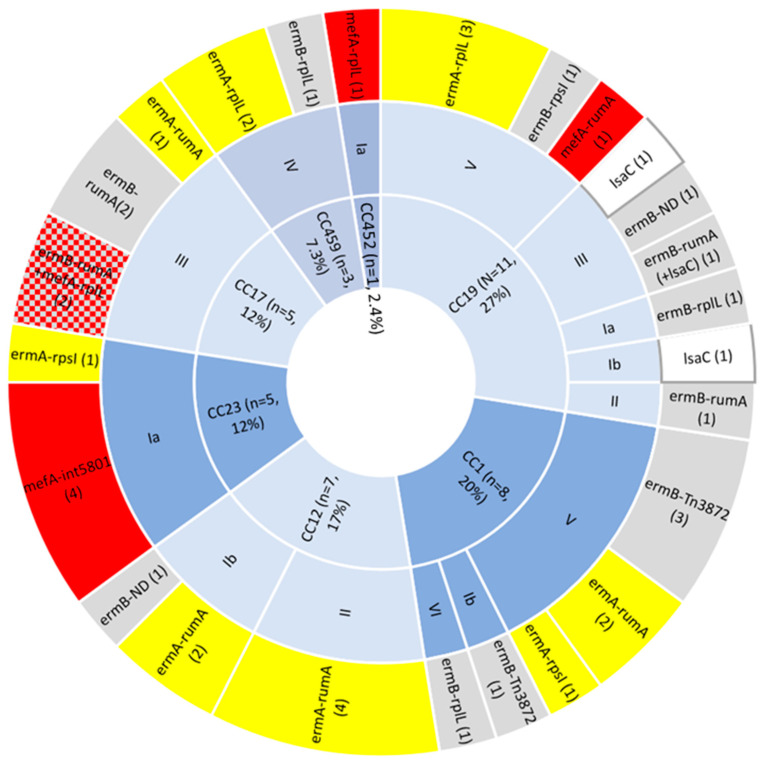
Distribution of macrolide and lincosamide resistance genes found in 41 of the 193 GBS isolates. Inner ring clonal clade frequency (number, percent of resistant isolates), middle ring serotype breakdown for clonal clade, outer ring macrolide/lincosamide resistance genes (colour coded: *lsa(C)* white, *erm(A)* yellow, *erm(B)* grey, *mef(A)* red, and *erm(B)*/*mef(A)* red and grey), noting associated mobile elements or site of genomic insertion; ND: Not determined. The *lsa(C)* element does not have associated MGE that was identified in this study.

**Figure 3 antibiotics-12-00544-f003:**
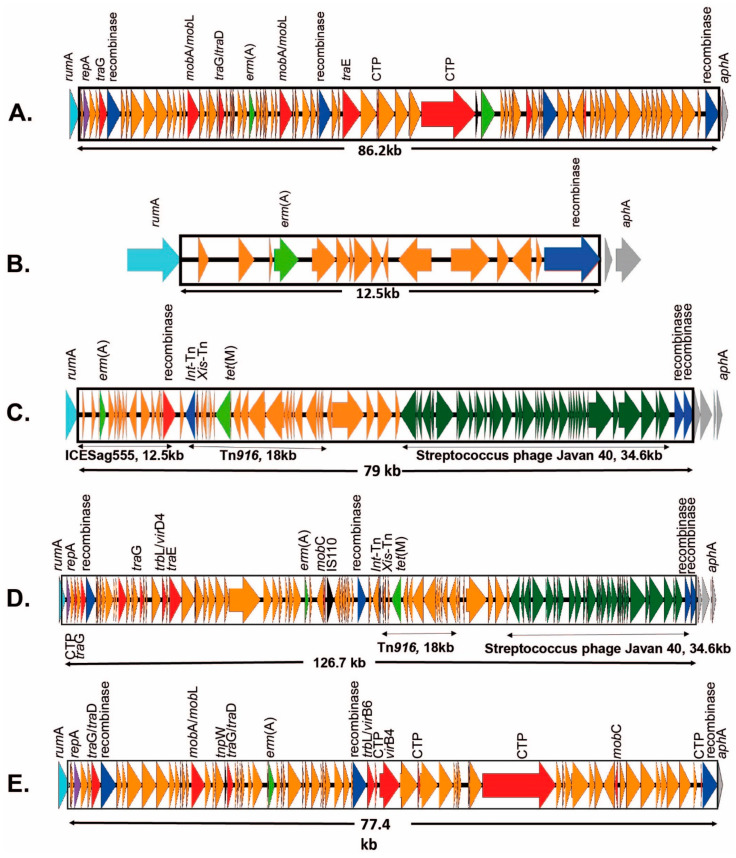
Novel ICE-carrying *erm(A)* genes inserted into the GBS chromosome at *rum*A gene (light blue). Arrows indicate gene orientation, antimicrobial resistance genes (light green), phage CDS (dark green), recombinase/integrase genes (dark blue), conjugal transfer proteins and other mobilising elements (red), replication initiator genes (purple), and other ICE genes (orange) relative to somatic post-insertion genes (grey); kb indicates the ICE size. (**A**) ICE-carrying *tet(M)* tn*916* elements for novel ICESag509, accession no. OP715843, found in four isolates. (**B**) ICESag066, accession no. OP715837, one isolate. (**C**) ICESag082, accession no. OP715838, which includes ICESag555 (found independently as a shorter ICE in a separate isolate). (**D**) ICESag098, accession no. OP715840, one isolate. (**E**) ICESag428, accession no. OP715841, one isolate.

**Figure 4 antibiotics-12-00544-f004:**
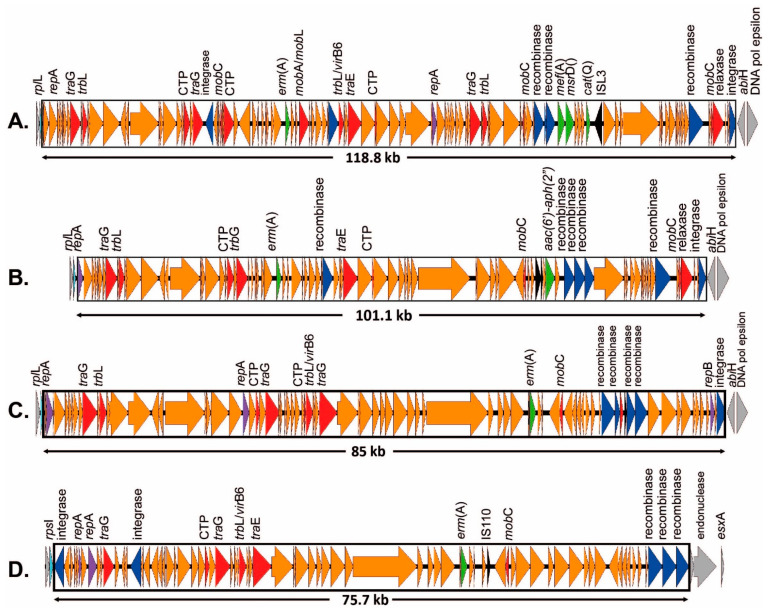
Novel ICE-carrying *erm(A)* genes inserted at *rplL* (**A**–**C**) or *rpsI* (**D**) genes. Orientation is shown by arrows for antimicrobial resistance genes (light green), recombinase/integrase genes (dark blue), conjugal transfer/mobilising elements (red), replication initiator genes (purple), and other ICE genes (orange) relative to conserved GBS genes (grey). (**A**) ICESag236 (accession no. OP508058, two isolates). (**B**) ICESag139 (accession no. OP508059, one isolate). (**C**) ICESag084 (accession no. OP715839, one isolate). (**D**) ICESag100414 (accession no. OP715842, one isolate).

**Figure 5 antibiotics-12-00544-f005:**
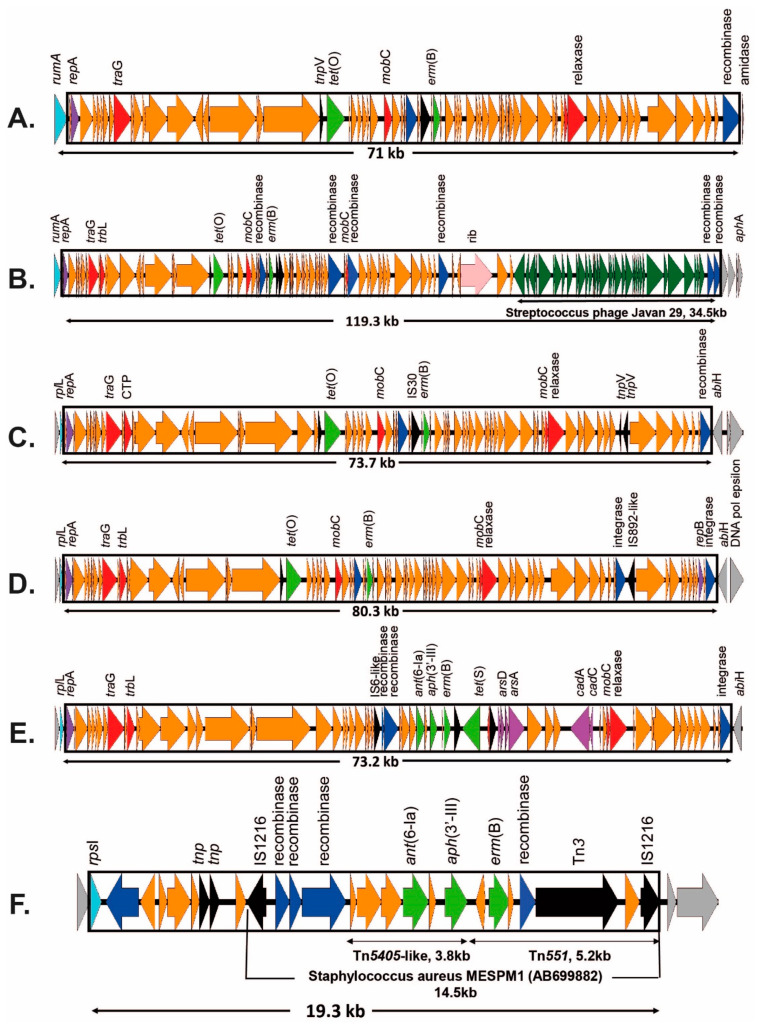
Novel ICE-carrying *erm(B)* genes inserted at *rum*A (**A**,**B**), *rplL* (**C**–**E**), or *rpsI* (**F**) genes. Orientation is shown by arrows for antimicrobial resistance genes (light green), recombinase/integrase genes (dark blue), conjugal transfer/mobilising elements (red), heavy metal resistance (purple), transposases (black), and other ICE genes (orange) relative to conserved GBS genes (grey). (**A**) ICESag624 (accession no. OP715844). (**B**) ICESag071 (accession no. OQ054582) (**C**). ICESag195 (accession no. OP508060) (**D**). ICESag206 (accession no. OP508061) (**E**). ICESag662 (accession no. OP508062). (**F**) ICESag738 (accession no. OP508063).

**Table 1 antibiotics-12-00544-t001:** Summary of GBS macrolide and lincosamide resistance genes by clonal complex and mobile genetic element.

Clonal Complex	Serotype	Insertion Site	Number of Isolates	ARGs	Mobile Genetic Element	Novel	Accession Number
CC1	Ib	random ^A^	1	*erm(B)* + *tet(M)*	Tn*3872*	N	OP715845
	V	random	3	*erm(B)* + *tet(M)*	Tn*3872*	N	OP715845
	V	*rps*I	1	*erm(A)*	ICESag100414	Y	OP715842
	V	*rum*A	2	*erm(A)* + *tet(M)* *erm(A)* + *tet(M)*	ICESag082 ICESag098	Y Y	OP715838 OP715840
	VI	*rpl*L	1	*erm(B)* + *ant*(6-Ia) + *aph*(3′-III) + *tet(S)*	ICESag662	Y	OP508062
CC12	Ib	*rum*A	2	*erm(A)* *erm(A)*	ICESag428 ICESag555	Y in ICESag082	OP715841 OP715841
	Ib	unknown ^B^	1	*erm(B)* + *tet(M)*	undefined	ND	none
	II	*rum*A	4	*erm(A)* + *tet(O)*	ICESag509	Y	OP715843
CC17	III	*rum*A	1	*erm(A)*	ICESag066	Y	OP715837
	III	*rum*A	2	*erm(B)* + *ant*(6-Ia) + *aph*(3′-III) + aad(E) + *tet(O)*	ICESag37	N	OP508056
	III	*rum*A + *rpl*L	2	*erm(B)* + *ant*(6-Ia) + *aph*(3′-III) + aad(E) + *tet(O)* + *mef(A)*/*msr(D)* ^C^	ICESag37 + ICESpy009	N	OP508056 + OP508057
CC19	II	*rum*A	1	*erm(B)* + *tet(O)*	ICESag624	Y	OP715844
	III	*rum*A (random)	1	*erm(B)* + *tet(M)* (+ *lsa*(*C*))	ICESag071 (+Tn*7589*)	Y Y	OQ054582 OQ269476
	Ia	*rpl*L	1	*erm(B)* + *tet(O)*	ICESag195	Y	OP508060
	Ib	unknown ^B^	1	*lsa(C)* ^D^	ICE2603tRNA^lys^	N	none
	III	unknown ^B^	1	*lsa(C)* ^D^	ICE2603tRNA^lys^	N	none
	III	unknown ^B^	1	*erm(B)* + *ant*(6-Ia) + *aph*(3′-III)	undefined	ND	none
	V	*rpl*L	1	*erm(A)* + *aac*(6′)-*aph*(2″)	ICESag139	Y	OP508059
	V	*rpl*L	2	*erm(A)* + *cat(Q)* + *mef(A)* + *msr(D)*	ICESag236	N	OP508058
	V	*rps*I	1	*erm(B)* + *ant*(6-Ia) + *aph*(3′-III)	ICESag738	Y	OP508063
	V	*rum*A	1	*mef(A)*/*msr(D)*	not apparent ^E^	ND	none
CC23	Ia	*rps*I	1	*erm(A)*	incomplete ^F^	ND	none
	Ia	Tn*5801*	4	*mef(A)*/*msr(D)* adjacent to *tet(M)*	not apparent	ND	none
	Ia	plasmid	1	*erm(T)*	plasmid	N	NZ_JF308630
CC459	IV	unknown	1	*erm(A)* + *tet(M)*	undefined	ND	none
	IV	*rpl*L	2	*erm(A)* *erm(B)* + *tet(O)*	ICESag084 ICESag206	Y Y	OP715839 OP508061
CC452	Ia	*rpl*L	1	*mef(A)*/*msr(D)*	ICESpy009	N	OP508061

Legend: ^A^ Tn*3872* elements are not directed by site-specific transposases, but are located in the same genomic location [21]. ^B^ Isolates, where the assembly contigs were quite short to identify the insertion sites, are listed as unknown. ^C^ Four isolates were identified with an identical insertion of ICESag37, two of which had an identical additional insertion of ICESpy009 at a different location [21]. ^D^ The two isolates with *lsa(C)* genes are located in ICE_2603_tRNA^lys^ elements, as described elsewhere [22]. ^E^ The *mef(A)*/*msr(D)* genes were located between the *rum*A gene and full Javan69 phage sequence (accession no. MK448828), but no mobile elements carrying these genes could be definitively defined. ^F^ The *erm(A)* gene was found to be inserted at the *rps*I gene located on the same assembly contig; however, the contig ended prior to identifying the 3′ end of the insertion; ND: Not determined.

## Data Availability

Genomic data are available according to the accession numbers provided in the text. Other data are available from the authors upon reasonable request.

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
