# Peer review of "Genomic Analysis Reveals New Integrative Conjugal Elements and Transposons in GBS Conferring Antimicrobial Resistance"

_antibiotics, 2023, doi:10.3390/antibiotics12030544_

Round 1
Reviewer 1 Report
The manuscript entitled, “Genomic analysis reveals new Integrative Conjugal Elements and transposons in GBS conferring antimicrobial resistance” will benefit the researchers working on Streptococcus agalactiae or Group B Streptococcus (GBS) and antimicrobial resistance. The manuscript needs a few amendments before considering for publication in Antibiotics. Therefore, minor revision is recommended to include the suggested changes. But personally, I enjoyed reading the manuscript and the suggestions are as follows:
Line 54: Which countries?
Line 75: Add ‘Of which,’ before 70.5%.......
Figure 1: Please expand or provide better quality.
Figure 2: Reduce the intensity of blue and grey colour. What does the red block mean, with no label? Better remove it.
Line 433-436 and 440: Add a recent reference for disk diffusion, consider https://doi.org/10.2478/s11756-020-00617-5.
Conclusion: Add a concluding statement on how the outcome will benefit future research/researchers.
Author Response
Response to reviewer 1
Thank you for your comments, we have improved our manuscript in line with your requests.
Line 54: Which countries?
>We have added the stated countries from the review referenced as “including Ireland, Italy, China, Portugal, Taiwan and the USA”
Line 75: Add ‘Of which,’ before 70.5%.......
>amended.
Figure 1: Please expand or provide better quality.
>A higher resolution Figure has been substituted for Figure 1.
Figure 2: Reduce the intensity of blue and grey colour. What does the red block mean, with no label? Better remove it.
>We have reduced the intensity of blue and grey colour as required -but maintained the borders between the different CC divisions. The text for the red block had been obscured by the size, but has been adjusted so that the label is now visible.
Line 433-436 and 440: Add a recent reference for disk diffusion, consider https://doi.org/10.2478/s11756-020-00617-5.
>We thank the reviewer for pointing out that a more recent reference is available for disk diffusion. We have updated this to the CLSI M32 2022 version of the document for both of these citation points and have updated reference 40 to a 2016 publication using D-zone inducible resistance for Group B Streptococcus clindamycin surveillance (https://doi.org/10.3205/dgkh000278 )
Conclusion: Add a concluding statement on how the outcome will benefit future research/researchers.
>We have added the following statement to the conclusion: “It was surprising to identify so many novel ICE present in a small cohort. The GenBank reference genomes for the mobile elements discovered here will be of benefit to future studies in determining whether increasing antimicrobial resistance is a result of global spread of the ICE defined here or whether they are being caused by sporadic capture of other novel ICE. The ICE sequences and genomic dataset provided by this and future studies enable development of tests to detect genes encoding resistance, improve surveillance and monitoring of resistance potential in the population and could aid more rapid therapeutic intervention than sensitivity testing alone.”
Reviewer 2 Report
Authors of the study investigated genetic and phenotypic relationships between antimicrobial resistance profiles of the Group B Streptococcus, which ended up validation of previously suggested regions and the discovery of new integrative conjugated elements novel transposons. The study was undertaken with care and expertise and the manuscript was written of great quality in terms of language, coherence and fluency. Authors are only suggested to extend conclusions with the potential use of the revealed information in human medicine.
Author Response
We thank the reviewer for their valuable input. As a similar request was asked by the other reviewer we have extended the conclusions with following statements “It was surprising to identify so many novel ICE present in a small cohort. The GenBank reference genomes for the mobile elements discovered here will be of benefit to future studies in determining whether increasing antimicrobial resistance is a result of global spread of the ICE defined here or whether they are being caused by sporadic capture of other novel ICE. The ICE sequences and genomic dataset provided by this and future studies enable development of tests to detect genes encoding resistance, improve surveillance and monitoring of resistance potential in the population and could aid more rapid therapeutic intervention than sensitivity testing alone.”